# Non-hermiticity in spintronics: oscillation death in coupled spintronic nano-oscillators through emerging exceptional points

Steffen Wittrock [1,2] ✉, Salvatore Perna[3], Romain Lebrun [1], Katia Ho [1], Roberta Dutra [4], Ricardo Ferreira [5], Paolo Bortolotti[1], Claudio Serpico[3] & Vincent Cros [1] ✉

The emergence of exceptional points (EPs) in the parameter space of a non-hermitian (2D) eigenvalue problem has long been interest in mathematical physics, however, only in the last decade entered the scope of experiments. In coupled systems, EPs give rise to unique physical phenomena, and enable the development of highly sensitive sensors. Here, we demonstrate at room temperature the emergence of EPs in coupled spintronic nanoscale oscillators and exploit the system's non-hermiticity. We observe amplitude death of self-oscillations and other complex dynamics, and develop a linearized non-hermitian model of the coupled spintronic system, which describes the main experimental features. The room temperature operation, and CMOS compatibility of our spintronic nanoscale oscillators means that they are ready to be employed in a variety of applications, such as field, current or rotation sensors, radiofrequeny and wireless devices, and in dedicated neuromorphic computing hardware. Furthermore, their unique and versatile properties, notably their large nonlinear behavior, open up unprecedented perspectives in experiments as well as in theory on the physics of exceptional points expanding to strongly nonlinear systems.

*Exceptional points* (EPs) are singularities in the parameter space of a system corresponding to the coalescence of two or more eigenvalues and the associated eigenvectors[1–6]. They are a peculiar feature of nonconservative (open) systems that have both loss and gain and they emerge when these two effects compensate. From the fundamental point of view, EPs play an important role in the area of non-Hermitian quantum theory based on $\mathcal{PT}$-symmetric Hamiltonians (with simultaneous parity-time invariance)[7]. In this context, they occur at phase transitions between broken-unbroken $\mathcal{PT}$-symmetry. While initially EPs were regarded as a mathematical-physics concept, in the last decade there has been a growing interest in EPs from the experimental point of view in such areas as atomic spectra measurements[8],

microwave cavity experiments[9–13], chaotic optical microcavities[14] or optomechanical systems[15]. Interestingly, exceptional points arise also in classical systems, such as coupled electric oscillators[16,17], optical systems[18], classical spin dynamics[19,20], and general dissipative classical systems[21].

Application-wise, the strong sensitiveness of eigenvalues to perturbations near EPs has been used to devise new types of sensors with unprecedented sensitivity[11,22,23]. This was demonstrated in highly sensitive optical nanoparticle detection[24,25], in laser gyroscopes[26], in optically pumped semiconductor rings for temperature detection[13], and in coupled microcantilevers for ultrasensitive mass sensing[27], however, none has been adapted to CMOS compatible systems.

[1]Laboratoire Albert Fert, CNRS, Thales, Université Paris-Saclay, 1 Avenue Augustin Fresnel, 91767 Palaiseau, France. [2]Helmholtz-Zentrum Berlin für Materialien und Energie GmbH, Hahn-Meitner-Platz 1, 14109 Berlin, Germany. [3]Department of Electrical Engineering and ICT, University of Naples Federico II, 80125 Naples, Italy. [4]Centro Brasileiro de Pesquisas Físicas (CBPF), Rua Dr. Xavier Sigaud 150, Rio de Janeiro 22290-180, Brazil. [5]International Iberian Nanotechnology Laboratory (INL), 471531 Braga, Portugal. ✉e-mail: steffen.wittrock@helmholtz-berlin.de; vincent.cros@cnrs-thales.fr

In the realm of magnetism, a field with profound implications for modern information technology, there has been growing attention on EPs and the effects of non-hermiticity in the last couple of years. Different approaches have been used exploiting the coupling of magnons to other distinct quantum systems, such as phonons[28] or explicitly photons[29–32], where great advance could be achieved in the field of cavity-magnonics[33]. In spintronics, the interest on EPs started with theoretical works on classical spin dynamics[19,20,34–38] and lately also includes spin wave physics[39,40]. Recent theoretical studies identify EPs as signatures of dynamical phase transitions relating linear and non-linear spin dynamics in their proximity[38], or find nontrivial non-hermitian topological phases in large arrays of coupled spintronic oscillators[41,42], or evidence complex bifurcations and bistability in a nonlinear coupled spintronic system[43] – aspects that are yet to be explored experimentally. It is to be noted that all works predominantly focus on local coupling mechanisms (except Ref. 43). Only recently, also based on a local coupling, the emergence of an EP could be demonstrated experimentally in magnonic $\mathcal{PT}$-symmetry devices[44]. However, the approach stays passive, i.e. the magnetic dissipation (damping) in two coupled magnetic layers is fixed and the system control (through the coupling) was realized through the choice of the thickness of the separating layer between the ferromagnetic thin films. The experimental study of EPs in coupled discrete nano-devices (with gain) has been so far overlooked and an experimental control of the coupled dynamics on a nonlocal circuit level is completely missing.

In this study, we demonstrate the presence of EPs in a system of two coupled spin-torque nano-oscillators (STNOs) (see Fig. 1) and how it can be used to control their oscillating state. STNOs are typical, CMOS-compatible, spintronic nanoscale devices in which the magnetization dynamics in a thin layer can be converted into electrical microwave signals[45]. They have both loss, associated with the damping of magnetization oscillations, and gain, provided by the transfer of spin angular momentum (through the spin torque effect) from a spin polarized current injected into the device (see Methods and Supplementary Information for further details). In this respect, coupled STNOs are archetypal non-hermitian systems to evidence the relevance of EPs in spintronics. We use STNOs based on the spin torque gyrotropic dynamics of a magnetic vortex core[45] (later on labelled as STVOs), since these show the best rf performance[46]. However, our results are applicable to all coupled STNO systems. We experimentally demonstrate that, by tuning the dc currents injected into the two STNOs, the position of an EP can be finely controlled leading to the

phenomenon of *amplitude death*[47]. This describes the vanishing of the oscillation amplitudes of the coupled STNOs, despite the increase of the spin torque (gain) in one oscillator. As shown later, such amplitude death occurs for a certain specific range of injected dc current values.

The presented results intend to create an important connection between the non-hermitian physics of EPs and spintronics, an area of research that has crucial technological implications for data storage and processing[48,49], sensor technology[48,50], wide-band high-frequency communications[51–56], and, more recently, bio-inspired networks for neuromorphic computing beyond CMOS[57,58]. Up to now, coupled STNOs have been studied mainly with respect to the phenomenon of mutual synchronization[59–63] and non-hermitian aspects are neglected or are about to be theoretically discovered[41,42]. Our observation provides the important experimental evidence to exploit EPs in coupled spintronics nano-devices.

## Results & discussion
### Theoretical modelling of EPs in spintronics

We first theoretically study the regime of small amplitude oscillations of the two coupled STVOs around their rest positions. From the linearized equations governing these small oscillations, the condition for an EP to emerge can be determined, leading to a formula that connects, at the EP, the relevant parameters of the coupled oscillators: frequencies, gain/loss parameters, and coupling coefficient. The position of an EP in the parameter space can hence be controlled in order to determine the interval of injected current values in which amplitude death occurs. The gain mechanism, counteracting the natural dissipation and enabling self-sustained oscillations in each STVO, is provided by the spin-transfer torque that is proportional to the injected dc current. Self-oscillations set in when the injected current $I$ is larger than a critical (threshold) current $I_c$, which corresponds to the exact compensation of gain and loss[61].

Gyrations of the vortex core around the symmetry axis of each oscillator are modeled by the Thiele-like theory for which the overall state of the oscillators is given by the in-plane displacements ($\boldsymbol{\rho}_1$, $\boldsymbol{\rho}_2$) of the vortex cores from the center of the corresponding devices (for details see Supplementary Information). The coupling between the two STVOs, which is assumed to be symmetric, is obtained by feeding strip-line antennas above each oscillator with the microwave voltage generated by the other, that in turn gives rise to a rf magnetic field (see Fig. 1). For vortex core displacements sufficiently small compared to the device radius, it is reasonable to assume a linear coupling[47,64] between the STVOs, reflecting the relevant range of the performed experiments. Importantly, the coupling has both dissipative and conservative terms that are described by the coefficients $k_d$ and $k_c$, respectively, and relate to the complex impedance of the electrical circuit.

The linearized Thiele equation governing the regime of small oscillations of the vortex cores around the rest position $\boldsymbol{\rho}_{1,2} = 0$, written in terms of the complex state variables $z_l = x_l + iy_l$ (with $l = 1, 2$) associated to the $l$-th vortex core $x$- and $y$-axis position, reads

$$\frac{d}{dt}\begin{bmatrix} z_1 \\ z_2 \end{bmatrix} = iA \cdot \begin{bmatrix} z_1 \\ z_2 \end{bmatrix}, \tag{1}$$

with

$$A = \begin{bmatrix} \omega_1 - i\beta_1 & k \\ k & \omega_2 - i\beta_2 \end{bmatrix}, \tag{2}$$

where $k = k_c - ik_d$, $\omega_l$ are the angular frequencies of vortex free oscillations, and $\beta_l$ are the loss/gain parameters. These latter parameters are given by $\beta_l = C_l I_l - d_l \omega_l$, where $d_l$ are the damping constants and $C_l$ are parameters determining the efficiency of the spin transfer effect, i.e. effectively $\beta_l > 0$ ($< 0$) corresponds to gain (loss).

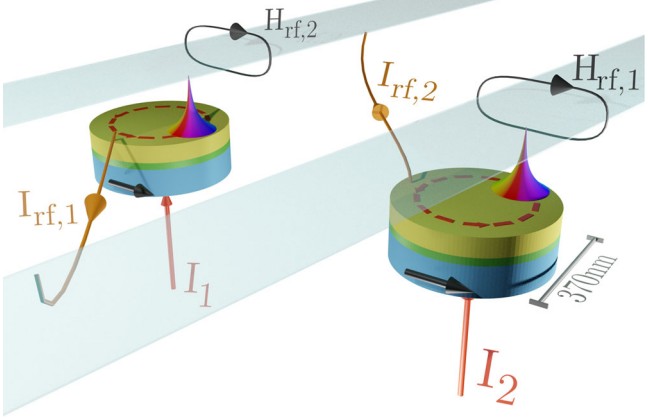

**Fig. 1 | Schematic view of the spintronic EP system made of coupled spintronic nano oscillators.** The coupling, which is designed to be symmetric, is obtained by feeding strip-line antennas above each oscillator with the microwave current $I_{rf}$ generated by the spin torque dynamics occurring in the other oscillator. This generates a magnetic rf field, $H_{rf}$, that hence generates the coupling[78]. The coupling is nonlocal and can be performed over long distances.

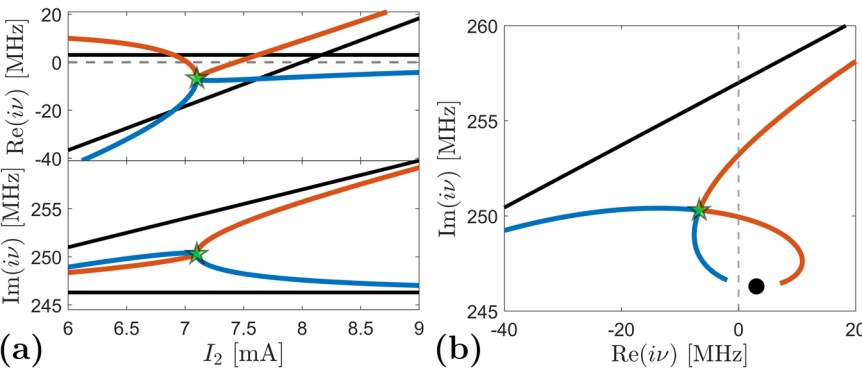

**Fig. 2 | Eigenvalues $i\nu_{1,2}$ from eq. (3) when the EP is placed at $I_1^* = I_{2,EP} = 7.1$ mA (green star). a** Real and imaginary part of the eigenvalues as a function of the dc current $I_2$. **b** Eigenvalues in the complex plane (Re($i\nu$), Im($i\nu$)). Black lines in (**a**) as well as black line and symbol in (**b**) refer to eigenvalues computed in the uncoupled case. Dashed gray lines depict the stability criterion (4): for Re($i\nu$) < 0, the rest position is stable and no auto-oscillations in the nonlinear sense occur. The system parameters can be found in the Supplementary Information. The coupling constant is $k_{EP} = 9.76 - 25.13i$.

The matrix $A$ in eq. (2) is non-hermitian and has indeed the typical form for systems exhibiting EPs[1,4]. In order to study the natural frequencies of the system (1), we assume a dependence of $z_{1,2}$ on the time of the type $e^{i\nu t}$. The natural frequencies $\nu_{1,2}$ are then obtained as eigenvalues of the matrix $A$ and they are given by the following formula:

$$\nu_{1,2} = \bar{\omega} - i\bar{\beta} \pm \sqrt{k^2 + \left(\tilde{\omega} - i\tilde{\beta}\right)^2}, \quad (3)$$

where

$$\bar{\beta} = (\beta_1 + \beta_2)/2, \quad \bar{\omega} = (\omega_1 + \omega_2)/2,$$
$$\tilde{\beta} = (\beta_1 - \beta_2)/2, \quad \tilde{\omega} = (\omega_1 - \omega_2)/2.$$

Stability of solutions is given by the following condition:

$$\text{Im}(\nu) \geq 0 \iff \text{Re}(i\nu) \leq 0. \quad (4)$$

By definition, EPs emerge when two natural frequencies coalesce along with the corresponding eigenvectors. This occurs when the square root term in eq. (3) is zero, leading to the following condition on the parameters to obtain an EP:

$$k_c - ik_d = \pm\left[\frac{(\beta_1 - \beta_2)}{2} + i\frac{(\omega_1 - \omega_2)}{2}\right], \quad (5)$$

where we have explicitly expressed $k, \tilde{\omega}$ and $\tilde{\beta}$.

We consider here the case that will be also presented in the experimental section: the current $I_1$ is fixed to a value $I_1^*$ and the second current $I_2$ is swept from values below to values above the threshold current $I_{c,2}$. By using the condition (5), the values of the parameters can be adjusted to have an EP at the desired value of the current $I_2$. The effect of the EP on the eigenvalues of the matrix $A$ is illustrated in Fig. 2. The black curves in Fig. 2a are the real and imaginary parts of the STVOs' eigenvalues when there is no coupling ($k_c = k_d = 0$). When coupling is taken into account, an EP exists at $I_{2,EP} = 7.1$ mA (green star), and it has the effect of attracting the two eigenvalues to one point in the complex plane. According to eq. (3), if, at the EP, $\bar{\beta} = (\beta_1 + \beta_2)/2$ is negative (this occurs when $\beta_2 < 0, |\beta_2| > \beta_1 > 0$), then both eigenvalues $i\nu$ have negative real parts and the system is lossy. This is also visible in Fig. 2b where the eigenvalues are plotted in the complex plane (Re($i\nu$), Im($i\nu$)): In the proximity of the EP, both eigenvalues are in the plane Re($i\nu$) < 0. This implies that the rest position of the two oscillators is stable, leading to the disappearance of both STVOs' oscillations in the nonlinear interpretation. This phenomenon is called *amplitude death* and, as we have illustrated above, it can be controlled by the

appropriate placing of the EP in the parameter space. Since the condition for the onset of an EP is very sensitive to perturbations, it might happen in experiments that the EP is not reached in a strict sense. Nevertheless, if the parameters are such that the condition (5) is nearly verified, the amplitude death phenomenon is expected to be reliably observed as well.

It is important to note that, when the rest position is stable in terms of the linearized model, we find that this stability is also exhibited by the rest position in the full nonlinear equations (see Supplementary Information). An important consequence is that, for predicting the phenomenon of amplitude death, the linear theory is strictly appropriate. On the other hand, when the real part of the eigenvalue $i\nu$ becomes positive – this happens when Re($i\nu$) crosses zero in Fig. 2 – the rest state becomes unstable. The regime that sets in after instability has an amplitude determined by the nonlinear saturation term in the Thiele equation and an approximate frequency of Im($i\nu$) at the aforementioned crossing. This phenomenon is referred to as a supercritical Andronov-Hopf bifurcation[65]. For values of parameters which correspond to a Hopf bifurcation point and no other bifurcations take place, the linear analysis can be used to estimate the frequency of the self-oscillating regimes by considering the imaginary parts of the natural frequency $i\nu$ at the Hopf bifurcation. In the assessment of the experimental results, this concept is applied in order to identify the appropriate parameter values describing the amplitude death region as a function of the injected currents.

## Experimental emergence of EPs

After having theoretically established the condition for the existence of an EP, we describe the experimental results that demonstrate the emergence of EPs and the correlated amplitude death regions in our coupled STVO system. All measurements have been conducted at room temperature. In the performed experiments, the current injected into the STVO 1 is kept constant to $I_1^*$, while sweeping the current $I_2$ injected into STVO 2. Note that the onset for self-sustained oscillations in the uncoupled case is $I_{c,1} \approx 6.95$ mA and $I_{c,2} \approx 8$ mA for STVO 1 and 2, respectively. The frequency evolution of the uncoupled STVOs with the applied current is similar (see Supplementary Information).

In Fig. 3a, we display the frequency spectra measured at $I_1^* = 8$ mA while $I_2$ is changed. For these conditions, no amplitude death is observed, however, a square-root-like frequency branching for $I_2 \geq 8$ mA is present. We ascribe this phenomenon to the presence of an EP in the linearized model. Therefore, we use formula (5) collocating the EP at $(I_{1,EP}^*; I_{2,EP}) = (8; 8)$ mA. Based on this identification, the linear coupling constant can be determined and the theory parameters adjusted in order to compute the eigenvalues $i\nu_{1,2}$ as a function of the current $I_2$ (Fig. 3b). From the general point of view, the eigenvalues $i\nu_{1,2}$

give information about the linear dynamics around the rest position. Their imaginary parts show a branching similar to the experimentally observed oscillation frequencies (Fig. 3a). Indeed, the theoretical linear approach provides a good access to the analysis of the intrinsically nonlinear regime of the experimental self-oscillations. In the Supplementary Information, we show the consistency of the linear model with numerical computations of the mutually coupled nonlinear dynamics based on the Thiele equations. Note that in a strict sense, the eigenvalues in the coupled system cannot be directly assigned to the single STVOs, but the modes must be regarded collectively, referring to the system matrix $A$. However, throughout the article we label the experimental signals corresponding to the STVO in which the oscillation mode is mainly localized, as corroborated by the nonlinear simulations and the theoretical analysis of the system eigenvectors under relatively small coupling (see Supplementary Information). In Fig. 3b, the real part confirms that the rest-position is unstable over the entire range $I_2$, i.e. $\mathrm{Re}(i\nu) > 0$, and hence self-oscillations are stabilized. The decrease of $\mathrm{Re}(i\nu)$ in the proximity of the EP is consistent with the experimental linewidth broadening in the range $I_2 \in [7.2; 8]$ mA in Fig. 3a, where noise-induced fluctuations become important due to the vicinity of the stability axis ($\mathrm{Re}(i\nu) = 0$).

From the prediction of our model, we expect that the decreasing of the gain effect (through the adjustment of the spin transfer torque in our case) together with the attraction of the eigenvalues around the

EP will make the amplitude death phenomenon observable. To confirm this behavior, in Fig. 4a–d, we perform measurements of the coupled system for smaller $I_1^*$ for which the eigenvalue real part can explicitly become negative due to the EP and hence, amplitude death occurs in this regime. With respect to the critical currents of the uncoupled STVOs, STVO 1 is undercritical in Fig. 4a and overcritical in Fig. 4b–d. The overall range of oscillation death evolves with $I_1^*$ (Fig. 4b–d), whereas rather the smaller value $I_2$ defining the amplitude death interval is affected than the larger one which remains quasi-constant. Increasing $I_1^*$ tends to stabilize the oscillation of STVO 1 and in consequence, counteracts the occurrence of the amplitude death. This leads to a decrease of the current range in which no oscillation is detected (see Fig. 4). Furthermore, for $I_1^* < 7.8$ mA and $I_2 > 8$ mA, the oscillations from STVO 1 show a lower output power together with a larger linewidth than it would be expected for self-sustained oscillations. When the current $I_1^*$ is close to 8 mA, in the vicinity of the EP, thermal noise can induce stochastic transitions between the oscillatory regime and the rest state corresponding to amplitude death (clearly visible in Fig. 4d). For currents $I_1^* \gtrsim 8$ mA (see Fig. 3a for $I_1^* = 8$ mA), oscillation death is no more occurring, however, the linewidth of the oscillation is clearly enhanced in a small range $I_2 \in [7; 8]$ mA. This range however decreases with increasing currents $I_1^*$. At even larger currents $I_1^* \gtrsim 9$ mA (see Supplementary Information), the two STVOs tend to mutually synchronize, a phenomenon that is commonly known for

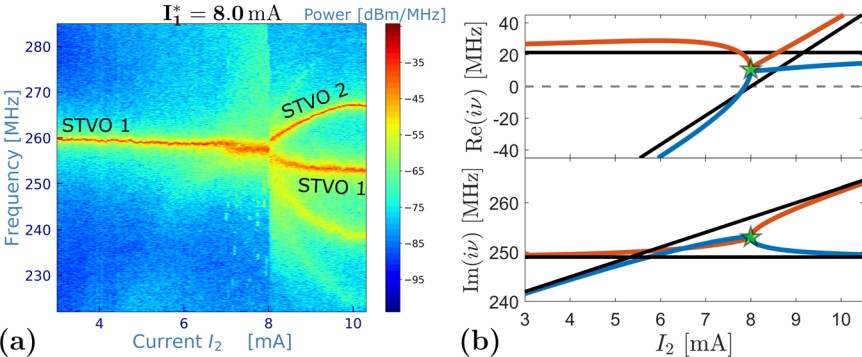

**Fig. 3 | Measured frequency spectra of the coupled system vs. current $I_2$ for $I_1^* = 8$ mA. a** The labeling corresponds to the STVO in which the oscillation mode is mainly localized. **b** Corresponding theoretically determined evolution of eigenvalues exhibiting an EP (green star). System parameters in Supplementary Information. The resulting coupling constant: $k_{EP} = 10.68 - 25.13i$.

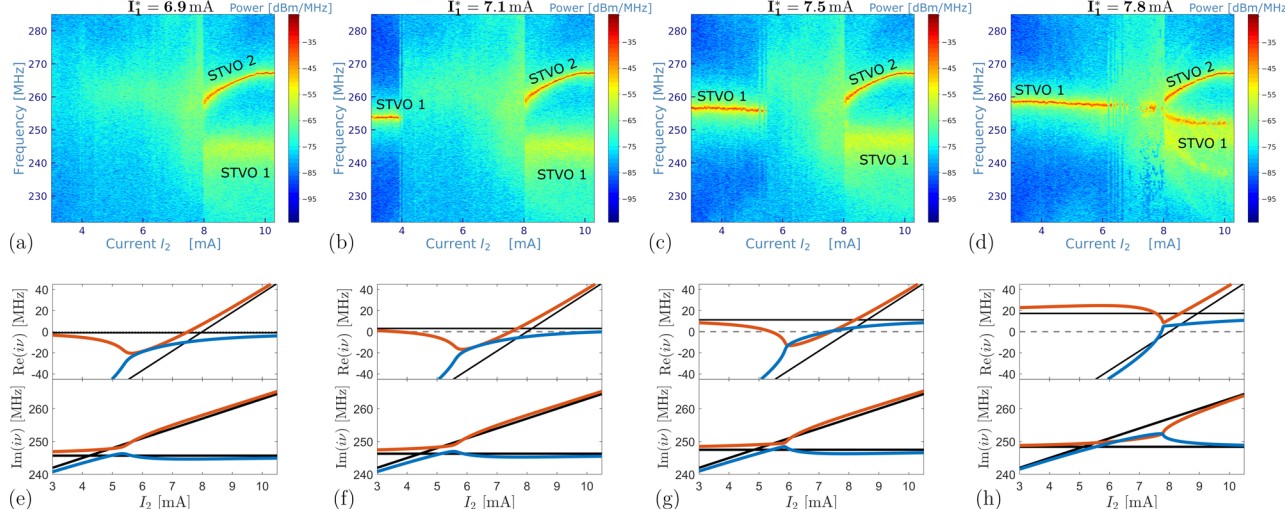

**Fig. 4 | Amplitude death and stochastic stability in the vicinity of the exceptional point.** Measured frequency spectra of the coupled system vs. current $I_2$ in STVO 2 for different currents $I_1^*$ in STVO 1 (**a–d**). **e–h** Real and imaginary part of $i\nu_1$ (red) and $i\nu_2$ (blue) fixing current $I_1$ and changing current $I_2$. Black lines refer to the same quantity evaluated when the coupling constant is set to 0. Experimental and theoretical graphs in the same column correspond to the same parameters.

STNOs[59–63] and which refers to the strongly nonlinear characteristics of the oscillator, far from the Hopf bifurcation point.

The experimentally observed amplitude death is very well reproduced by our modelling of the coupled STVOs. In Fig. 4e–h, we present the corresponding real and imaginary parts of the eigenvalues $iv_{1,2}$ as a function of the current $I_2$. Except for the value of the coupling constant, which in principle depends on the electric interface between the STVOs as well as on their dynamical state, the modelling parameters are the same as those used in Fig. 3b. We find that by only rotating the before determined coupling constant $k_{EP}^* \rightarrow k_{EP}^* e^{i\phi_k}$ in the complex plane $(k_c, k_d)$ without changing its modulus, the amplitude death phenomena can be completely described. The rotation angle for the two cases where the amplitude death is evident at $I_1^* = 7.1$ and 7.5 mA (Fig. 4b, c) is $\phi_k = 40$ and 45°, respectively. In Fig. 4f–g, the amplitude death current ranges can be recognized by looking at where the condition $Re(iv_{1,2}) \leq 0$ is satisfied. Then, at the upper current value $I_2$ of the amplitude death regime, the real part of one eigenvalue crosses the real axis and the corresponding mode becomes unstable. This situation corresponds to a Hopf bifurcation which brings the system to self-oscillations. Such consideration permits to rigorously justify the presence of the upper branch in the measured spectra. The discussed Hopf bifurcation point does not significantly change its position while the square root like upper branch of $Re(iv)$ at lower currents $I_2$ implies a strong dependence of the amplitude death range's lower boundary on the fixed current $I_1^*$, as also found experimentally. For larger current $I_2$, in the case $I_1^* = 7.5$ mA, also the real part of the other eigenvalue (blue curve in $Re(iv)$) becomes positive, but it stays close to the real axis. In the experiments, which are subject to thermal fluctuations, this manifests as the described linewidth broadening of STVO 1's oscillation at relative smaller power. Similar situation occurs for $I_1^* = 6.9$ mA and $I_1^* = 7.1$ mA. In both cases the value of the rotation angle is set to $\phi_k = 40°$. The main difference with the $I_1^* = 7.5$ mA case is that only the real part of one eigenvalue crosses the real axis. The other stays close to it. Similar to before, thermal fluctuations shall permit oscillations, however exhibiting a large linewidth in the experiments. For $I_1^* = 7.8$ mA, the measured spectra are similar as for $I_1^* = 8$ mA and hence we set $\phi_k = 0°$. The oscillations' death for this case is experimentally observed (see Fig. 4d), but is not described by the linear theory. Note that nonlinearity might become more important in this regime. However, the stochasticity of the transitions between oscillation regime and rest state suggests that also thermal fluctuations play in this case a dominant role in determining the stability of the oscillators.

Indeed, the main characteristics of the coupled system can be accessed by the developed linearized theory. The study of the eigenvalues as a function of the current permits to unravel the key features of the coupled STVO system's frequency response.

In conclusion, we exploit the non-hermiticity of two coupled spintronic nano-oscillators and demonstrate the emergence of EPs in this spintronic system, which is in fact promising candidate for multiple potential applications[45]. The existence of an EP drastically influences the eigenvalue characteristics leading to various complex phenomena, such as oscillation death or stochastic oscillation stability. We develop a theoretical modelling and show that the main experimental features at this stage can be well reproduced by linearized coupled spintronic equations.

## Outlook

One of the interesting specificities of the spintronic nano-oscillators is their strong nonlinearity which makes them a promising candidate for various applications and leads to a tremendous manifold of physical phenomena unified in these nanoscale devices. The emergence of an EP in a nanoscale nonlinear system is to our opinion of fundamental interest. Beyond the already mentioned implications for the development of novel types of spintronic sensors operating at exceptional points[11,22], these systems are anticipated to unravel fascinating physics. This includes phenomena such as chaos, complex bifurcations[43], or the emergence of topological operations around the EP[15,66]. Complex dynamics and as well the demonstrated occurrence of stochastic stability might furthermore complement the field of hardware-based neuromorphic computing that recently gained attention in the context of spintronics[67], for instance as stochastic spiking neurons. Non-hermiticity in this respect adds an additional complex response of the system to input signals[68–70], implying abrupt phase transitions which are also inherent in neural networks[71]. Characteristics of non-hermiticity have been found in the description of the brain, for instance in EEG measurements[72,73], or the inhibitory and excitatory balance in neocortical neurons[74], similar to nonconservative elements of gain and loss in our STNO system. We emphasize that higher dimensionally coupled systems have been realized with STNOs[63,75,76] which are anticipated to facilitate the emergence of higher order exceptional points[13,36,40] or other complex dynamics[41,42]. All these different aspects are still to be explored and potentially lead to intriguing findings in nonlinear non-Hermitian systems on the nanoscale.

## Methods

More extensive information along with additional data and discussion regarding theory, simulations, and experiment can be found in the supplementary information file.

### Device fabrication

The studied STVO devices are magnetic tunnel junctions containing a pinned layer made of a conventional synthetic antiferromagnetic stack (SAF), a MgO tunnel barrier and a NiFe-free layer in a magnetic vortex configuration (blue, green and yellow layers in Fig. 1, resp.). The magnetoresistive ratio related to the tunnel magnetoresistance effect (TMR) lies around 110% at room temperature and the area resistance product is $RA \approx 2\,\Omega\mu m^2$. In detail, the SAF is composed of IrMn(60)/$Co_{70}Fe_{30}$(2.6)/Ru(0.85)/$Co_{40}Fe_{40}B_{20}$(2.6) and the total layer stack is Ta(5)/CuN(50)/Ta(5)/Ru(5)/SAF/MgO(1)/$Co_{40}Fe_{40}B_{20}$(2)/Ta(0.2)/$Ni_{80}Fe_{20}$(7)/Ta(10)/CuN(30)/Ru(5), with the nanometer layer thickness in brackets. The growth of the amorphous NiFe-free layer is decoupled from the lower CoFeB layer by a 0.2 nm Ta-layer. This structure permits to exploit the high tunnel magnetoresistance (TMR) ratio of the crystalline CoFeB-junction and the magnetically softer NiFe for the vortex dynamics. The layers are deposited on high resistivity $SiO_2$ substrates by ultrahigh vacuum magnetron sputtering and subsequently annealed for 2 h at $T = 330\,°C$ at an applied magnetic field of 1 T along the SAF's easy axis. The patterning of the circular tunnel junctions is conducted using e-beam lithography and Ar ion etching. They have an actual diameter of $2R = 370$ nm and the microwave field line of $1\,\mu m \times 300$ nm is lithographied 300 nm above the nanopillar.

## Data availability

The data generated in this study and supporting the manuscript experimental figures have been deposited in a Zenodo database under https://doi.org/10.5281/zenodo.10058698[77].

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

## Acknowledgements

S.W. acknowledges financial support from Labex FIRST-TF under contract number ANR-10-LABX-48-01. The work is supported by the French National Research Agency ANR project "SPINNET" ANR-18-CE24-0012 and "ICARUS" ANR-22-CE24-0008-01, and by a France 2030 government grant (ANR-22-EXSP-0005 PEPR SPIN-SPINCOM). K.H. acknowledges financial support from ANRT under contract number 2021/1341. The Horizon2020 Framework Program of the European Commission, under FET-Proactive Grant agreement No.899646 (k-NET) and the project number 101070287 – SWAN-on-chip – HORIZON–CL4-2021-DIGITAL-EMERGING are also acknowledged for support.

## Author contributions

S.W., R.L., C.S., and V.C. conceived the project. R.F. and R.D. prepared the devices. S.W. performed the experimental measurements and analyzed the data with the help of V.C., R.L., and P.B. C.S and S.P. developed the theoretical model with the help of S.W. S.P. and C.S. conducted the numerical simulations. S.W., V.C., R.L., C.S., and S.P. prepared the manuscript and all authors discussed and contributed to the final version.

## Funding

## Competing interests

The authors declare no competing Interests.
