## [Peer Review File · Nature Communications]

Reviewers' Comments:

Reviewer #1:

Remarks to the Author:

In this work, the authors investigate experimentally a system of coupled STNOs from a non-Hermitian perspective. Specifically, they explore the relation between the emergence of EPs in the linearized dynamics and dynamical phase transitions of the non-linear spin dynamics.

I think, upon few revisions, this work deserves publications as it contains the first experimental evidence to exploit EPs in coupled spintronics nano-devices.

Here few questions/suggestions that I would like the authors to address before recommending publication:

- i) Recent theoretical work [Phys. Rev. B 107, L100402 (2023)] has uncovered a direct relation between the emergence of EPs and dynamical phase transitions of the nonlinear spin dynamics and has also predicted that crossing an exceptional point in the coupled magnetization dynamics can lead to emergence (or suppression) of a dynamical oscillatory regime. This work is not cited by the authors but it seems quite prominent to their investigation.
- ii) Could the authors elaborate more on the physical origin of the real and imaginary coupling between the STNOs?
- iii) I think it would help the reader if the authors would explicitly state the sign for which $\beta_{\{l\}}$ corresponds to loss/gain
- iv) Along the same lines, it would help the reader if the authors could state explicitly whether the imaginary part of the natural frequencies in fig. 2 correspond to gain / loss.
- v) Can the authors specify which type of Hopf bifurcation they are discussing? A supercritical one?

Reviewer #2:

Remarks to the Author:

Non-Hermitian physics with PT symmetry has recently prospered in a large variety of research areas such as photonics, electronics, superconductivity, etc. In particular, PT symmetry in magnetism has been extensively explored theoretically, however, only one experimental result of PT symmetry with observable exceptional point has been reported. In that work, the balanced magnetic loss and gain in PT symmetry has been replaced with different magnetic losses. Up to now, the coupled magnetic loss and gain has not been achieved yet. In this manuscript, the authors advanced magnetic PT symmetry with realizing coupled magnetic loss and gain by spin transfer torque in two magnetic vortex structures. The exceptional point has been reached by increasing the current intensities and amplitude death of self-oscillation has been observed. The work is of great importance and will certainly attract great attentions for spintronics and quantum physics research communities. The manuscript has been well written and organized. Based on the above reasons, I recommend its publication as it is.

Reviewer #3:

None

Answer to the reviewers

Reviewer comment in black

Answer in blue-gray

First, we would like to thank the referees for their reports, the time they spent on our work, and the points they have raised. We will address in the following all the comments in order to improve our manuscript.

Reviewer Comments:

Reviewer 1

In this work, the authors investigate experimentally a system of coupled STNOs from a non-Hermitian perspective. Specifically, they explore the relation between the emergence of EPs in the linearized dynamics and dynamical phase transitions of the non-linear spin dynamics.

I think, upon few revisions, this work deserves publications as it contains the first experimental evidence to exploit EPs in coupled spintronics nano-devices.

Here few questions/suggestions that I would like the authors to address before recommending publication:

i) Recent theoretical work [Phys. Rev. B 107, L100402 (2023)] has uncovered a direct relation between the emergence of EPs and dynamical phase transitions of the nonlinear spin dynamics and has also predicted that crossing an exceptional point in the coupled magnetization dynamics can lead to emergence (or suppression) of a dynamical oscillatory regime. This work is not cited by the authors but it seems quite prominent to their investigation.

Thank you for drawing our attention on this very recent work. Indeed, we have become aware of that publication in the meantime and will now cite it in our revised manuscript version as it is absolutely pertinent.

The mentioned theoretical work is studying a coupled bilayer system with the bottom layer exhibiting a negative effective damping assumed to be related to spin injection. We would like to mention that this problem is indeed very similar to our model from the mathematical perspective. However, from the physical point of view it is different because the work treats coupled macrospins rather than coupled vortices.

In the manuscript, we rephrase the following part in the introduction (II. 71-79): *“Very recently, EPs have been theoretically described as signatures of dynamical phase transitions relating linear and nonlinear spin dynamics in their proximity [41]. In coupled STNOs, the potential of non-hermitian effects has been theoretically emphasized in larger dimensional arrays, in which nontrivial non-hermitian topological phases can emerge[44, 45], an aspect that is yet to be explored experimentally, or predicted to facilitate large amplitude oscillations[41].”*

ii) Could the authors elaborate more on the physical origin of the real and imaginary coupling between the STNOs?

We have chosen the most general approach to realize a physical coupling scheme, that is to include both a real and an imaginary contribution to the coupling. Note that the rf fields themselves couple conservatively to the dynamics. However, in the experiment, both mechanisms will always be present due to the complex impedance of the electronic connection, with real (resistive) and imaginary (reactive) component. In the revised manuscript, we clarify this in ll. 147 – 150:

“Importantly, the coupling has both dissipative and conservative terms that are described by the coefficients k_d and k_c , respectively, and relate to the complex impedance of the electrical circuit.”

iii) I think it would help the reader if the authors would explicitly state the sign for which β_l corresponds to loss/gain

We have added an additional clarification in ll. 153 ff.: *“These latter parameters are given by $\beta_l = C_l I_l - d_l \omega_l$, where d_l are the damping constants and C_l are parameters determining the efficiency of the spin torque effect, i.e. effectively $\beta_l > 0$ (< 0) corresponds to gain (loss).”*

iv) Along the same lines, it would help the reader if the authors could state explicitly whether the imaginary part of the natural frequencies in fig. 2 correspond to gain / loss.

We specify in the caption of figure 2: *“[...] dashed gray lines depict the stability criterion (4).”* Note that in the vocabulary of linear dynamics, lossy/sustained dynamics is described by the mentioned supercritical Hopf bifurcation point, which is characterized by the stability criterion (4). Furthermore, in order to state this fact clearer, we add to the main text (ll. 170 ff.): *“According to eq. (3), if, at the EP, $\bar{\beta} = (\beta_1 + \beta_2)/2$ is negative (this occurs when $\beta_2 < 0, |\beta_2| > \beta_1 > 0$), then both eigenvalues ν have negative real parts and the system is lossy. This is also visible in fig. 2b where [...]”.*

v) Can the authors specify which type of Hopf bifurcation they are discussing? A supercritical one?

Yes, we are talking about a supercritical Andronov-Hopf bifurcation. In the revised manuscript, we now state the type of Hopf bifurcation clearly (l. 203).

Reviewer 2

Non-Hermitian physics with PT symmetry has recently prospered in a large variety of research areas such as photonics, electronics, superconductivity, etc. In particular, PT symmetry in magnetism has been extensively explored theoretically, however, only one experimental result of PT symmetry with observable exceptional point has been reported. In that work, the balanced magnetic loss and gain in PT symmetry has been replaced with different magnetic losses. Up to now, the coupled magnetic loss and gain has not been achieved yet. In this manuscript, the authors advanced magnetic PT symmetry with realizing coupled magnetic loss and gain by spin transfer torque in two magnetic vortex structures. The exceptional point has been reached by increasing the current intensities and amplitude death of self-oscillation has been observed. The work is of great importance and will certainly attract great attentions for spintronics and quantum physics research communities. The manuscript has been well written and organized. Based on the above reasons, I recommend its publication as it is.

We thank the reviewer for this very positive assessment.

Reviewers' Comments:

Reviewer #1:

Remarks to the Author:

The authors have addressed my comments in a satisfactory manner. I recommend publication as it is.

Answer to the reviewers

Reviewer comment in black

Answer in blue-gray

We thank Referee 1 and 2 for recommending publication of our work. Below we address some of their remaining comments which helped clarifying some points of our manuscript. Furthermore, we in detail respond to the criticism risen by Referee 3 about the perspectives opened by our work for the study of EPs in spintronic devices.

Reviewer Comments:

Reviewer 1

In this work, the authors investigate experimentally a system of coupled STNOs from a non-Hermitian perspective. Specifically, they explore the relation between the emergence of EPs in the linearized dynamics and dynamical phase transitions of the non-linear spin dynamics.

I think, upon few revisions, this work deserves publications as it contains the first experimental evidence to exploit EPs in coupled spintronics nano-devices.

We thank Referee 1 for pointing out our efforts to evidence that for the first time, EPs can be stabilized and manipulated in active CMOS compatible spintronic devices. Below we address his/her remaining questions:

Here few questions/suggestions that I would like the authors to address before recommending publication:

i) Recent theoretical work [Phys. Rev. B 107, L100402 (2023)] has uncovered a direct relation between the emergence of EPs and dynamical phase transitions of the nonlinear spin dynamics and has also predicted that crossing an exceptional point in the coupled magnetization dynamics can lead to emergence (or suppression) of a dynamical oscillatory regime. This work is not cited by the authors but it seems quite prominent to their investigation.

Thank you for drawing our attention on this very recent work. Indeed, we have become aware of that publication in the meantime and will cite it in our revised manuscript version as it is definitely pertinent to our work.

In this theoretical work, the authors are studying a coupled bilayer system with the bottom layer exhibiting a negative effective damping assumed to be related to spin current injection. We would like to mention that this problem is indeed very similar to our model from the mathematical perspective. However, from the physical point of view, it treats the case of coupled macrospins (uniform magnetization) rather than coupled vortices like in our case. The similarity promisingly highlights that EPs are a general feature of STNOs, as we also point out in our manuscript, and that their experimental demonstration is of importance for a large audience in condensed matter physics.

In the revised manuscript, we rephrase the following part in the introduction (ll. 71-79): *“Very recently, EPs have been theoretically described as signatures of dynamical phase transitions relating linear and nonlinear spin dynamics in their proximity [41]. In coupled STNOs, the potential of non-hermitian effects has been theoretically emphasized in larger dimensional arrays, in which nontrivial non-hermitian topological phases can emerge [44, 45], an aspect that is yet to be explored experimentally, or predicted to facilitate large amplitude oscillations[41].”*

ii) Could the authors elaborate more on the physical origin of the real and imaginary coupling between the STNOs?

On one hand, for the theoretical modelling, we have chosen the most general approach to realize a physical coupling scheme, that is to include both a real and an imaginary contribution to the coupling. On the other hand, in the experiment, both mechanisms are always present due to the complex impedance of the electronic connection, with real (resistive) and imaginary (reactive) components. In the revised manuscript, we clarify this point in ll. 147 – 150: *“Importantly, the coupling has both dissipative and conservative terms that are described by the coefficients k_d and k_c , respectively, and relate to the complex impedance of the electrical circuit.”*

iii) I think it would help the reader if the authors would explicitly state the sign for which β_l corresponds to loss/gain

We have added an additional clarification in ll. 153 ff.: *“These latter parameters are given by $\beta_l = C_l I_l - d_l \omega_l$, where d_l are the damping constants and C_l are parameters determining the efficiency of the spin transfer effect, i.e. effectively $\beta_l > 0$ (< 0) corresponds to gain (loss).”*

iv) Along the same lines, it would help the reader if the authors could state explicitly whether the imaginary part of the natural frequencies in fig. 2 correspond to gain / loss.

In order to state this point clearer, we specify in the caption of figure 2: *“[...] Dashed gray lines depict the stability criterion (4): for $Re(iv) < 0$, the rest position is stable and no auto-oscillations in the nonlinear sense occur.”*

Note that in the vocabulary of linear dynamics, lossy/sustained dynamics is only described by the mentioned supercritical Hopf bifurcation point, which is characterized by the stability criterion (4). In the strict sense, sustained dynamics (i.e. instability in the linear sense) is a nonlinear term, as elaborated in ll. 190 – 202: *“[...] On the other hand, when the real part of the eigenvalue iv becomes positive – this happens when $Re(iv)$ crosses zero in fig. 2 – the rest state becomes unstable. The regime that sets in after instability has an amplitude determined by the nonlinear saturation term in the Thiele equation and an approximate frequency of $Im(iv)$ [...]”*.

Moreover, to be clearer, we add in the revised version of the main text (ll. 170 ff.): *“According to eq. (3), if, at the EP, $\bar{\beta} = (\beta_1 + \beta_2)/2$ is negative (this occurs when $\beta_2 < 0, |\beta_2| > \beta_1 > 0$), then both eigenvalues iv have negative real parts and the system is lossy. This is also visible in fig. 2b where [...]”*.

v) Can the authors specify which type of Hopf bifurcation they are discussing? A supercritical one?

Yes, we are talking about a supercritical Andronov-Hopf bifurcation. In the revised manuscript, we more explicitly state the type of Hopf bifurcation clearly (l. 202).

Reviewer 2

Non-Hermitian physics with PT symmetry has recently prospered in a large variety of research areas such as photonics, electronics, superconductivity, etc. In particular, PT symmetry in magnetism has been extensively explored theoretically, however, only one experimental result of PT symmetry with observable exceptional point has been reported. In that work, the balanced magnetic loss and gain in PT symmetry has been replaced with different magnetic losses. Up to now, the coupled magnetic loss and gain has not been achieved yet. In this manuscript, the authors advanced magnetic PT symmetry with realizing coupled magnetic loss and gain by spin transfer torque in two magnetic vortex structures. The exceptional point has been reached by increasing the current intensities and amplitude death of self-oscillation has been observed. The work is of great importance and will certainly attract great attentions for spintronics and quantum physics research communities. The manuscript has been well written and organized. Based on the above reasons, I recommend its publication as it is.

We thank the reviewer for this very positive assessment pointing out the novelty and the importance of our work.

Reviewer 3

The manuscript under review presents the observation of an exceptional point (EP) in a system consisting of two coupled vortex-based spin-torque nano-oscillators (STVOs). It is based on modeling this system theoretically, claiming oscillation death, "frequency branching", and stochastic oscillation

as outcomes of the EP in this system. The authors also propose the potential applicability of the EP, asserting its ability to enhance sensitivity and its potential use in nonconventional computing and logic.

While the fundamental interest in EP is apparent and timely, I question the fit of this manuscript for the readership of Nature Nanotechnology. The study's claim of potential applicability appears tentative, as it only demonstrates possible oscillation death, without providing clear suggestions for its implementation in sensors or other applications, the presence of EPs in nano-oscillators, yet to be proven by the authors, might be expected. Absent a unique feature or a clear path to utility, I'm unsure how this study aligns with the standards of Nature Nanotechnology, taking also into account that the linear model developed by the authors appears to be restricted to STVOs, at a low amplitude, potentially limiting the broader applicability of the model. where I struggle to see how this linear model could account for spintronic oscillators that are mainly characterized and attractive through their nonlinearity. So despite the manuscript claims to pioneering the study of non-hermitian phenomena in coupled spintronic oscillators, I believe the manuscript's scope might be better suited for a specialized journal such as Phys. Rev. B, or possibly a broader journal such as Phys. Rev. Applied, given certain issues, are addressed more properly.

We understand that the referee worries a lot about the standards of **Nature Nanotechnology**, which we could have understood in case our article would have been submitted for this Nature family journal, **BUT** this is not the case, and this manuscript is under evaluation in **Nature Communications**, with very positive conclusion from reviewers 1 and 2.

Note that it does not mean that we are not interested in possible applications of our results. In that sense, the potential applicability that we put forward in the manuscript is very similar to those also indicated by other groups investigating EPs in different experimental systems. The advantage compared to them, and even if we have not yet reached this level of development, is that this first demonstration of EPs using spintronic devices opens opportunities to apply this physics in nanoscale, CMOS compatible and already commercially available devices. To our opinion, this is a strong statement for the future impact of our work.

Another possible confusion of the reviewer that we want to answer straightforwardly is that our study is not only theoretical, but also presents some detailed experimental data, showing some features that, thanks to the excellent agreement with our theoretical modeling, we ascribed to the presence of an EP in our system based on coupled spintronic oscillators.

In the following, we clarify more specifically the novelty of work, among which, why it represents a first step towards future observations and implementation of EPs in spintronic devices. All of them makes that first, Nature Communications is the right scope for our article and second, that the standards of this journal are achieved.

- **EPs in a coupled STNO system:** First, claiming that STNOs are only interesting due to their nonlinearity is a statement made by the reviewer that we, and many others, do not share. First, as already mentioned, they are also nano-sized devices and can be integrated in CMOS compatible systems. Second, a lot of research has been performed in the recent years on using STNOs as rf rectifiers operating in the linear regime with a lot of interest for, e.g., rf detection or rf energy harvesting. Apart from that, STNOs exhibit many intriguing properties of which many, but not all, are related to their (strong) nonlinearity. We can thus notice that one of the advantages of STNOs compared to other physical systems is that their properties can be tuned from highly nonlinear to linear. These two regimes can, e.g., be specifically

exploited in neuromorphic spintronics (see for example, [Sethi et al., <https://arxiv.org/ftp/arxiv/papers/2301/2301.03794.pdf>]).

Second, a key message of our manuscript is that the presence of an EP in coupled nano-oscillators CAN be expected! However, this is not primarily obvious, especially in terms of the EP's "reachability" in terms of the system's parametrization and its real control parameters (such as dc current or magnetic field). It is thus to be noticed in fig. 2 that the range of stabilization is rather small and that it requires a fine tuning of the coupling parameters which is more easily achieved using electrical coupling as we originally do in this study than with standard dipolar coupling defined during a lithography process.

- **Modelling approach:** Our theoretical model indeed describes our specific STVO system. This is what it is supposed to do. The study of the linearized dynamics close to the Hopf bifurcation is a well-established approach and therefore, the reviewer's criticism could apply to a huge manifold of systems and most of previous works on EPs. We refer to standard textbooks of nonlinear dynamics and oscillators of which many exist, e.g. [Pikovsky, Rosenblum, Kurths: *Synchronization, a universal concept in nonlinear sciences. (2001)*]. Furthermore, we point out that the employed Thiele equation can also quite easily be generalized to, e.g., the Landau-Lifshitz-Gilbert equation or a Stuart-Landau equation and hence, another important message of our work is that the STVO system represents a model for other magnetic systems.
- **Applicability:** The applicability of the non-hermitian effect is so far purposely mentioned in the Outlook section, since the various potential applications are indeed yet to be demonstrated. However, various approaches are cited (e.g. Refs. [11, 22, 23] for non-hermitian sensing, Refs. [58, 75] for neuromorphic computing with coupled STNOs, or [68, 69, 70] for neuromorphic functionality of non-hermitian systems) and potential application approaches can be deduced from those citations, including both, applications of other non-hermitian systems and as well applications of coupled STNO systems. In this context, in the revised manuscript, we add one more pertinent reference in the outlook section [Yu et al., *Adv. Science* 6, 15, 1900771 (2019)] covering non-hermitian physics in neuromorphic systems.
Importantly, approaches for applications must and will be demonstrated based on our discoveries, which are the foundation for future works in this area.

Nevertheless, the authors might use the flexibility that the spintronic oscillators and the availability of off-shelf microwave components to add substantial experimental demonstration affirming their

1-There is no sufficient description of the samples probably a better picture or detailed description is needed, how much are the MTJs placed spatially is there any dipolar coupling between them, I assume that the Vortex is a ground state in those samples without any current, is there is any additional coupling, dynamic within the MTJ itself.

For a thorough description of the sample system, we refer to the Methods section 3. In addition to that, we add in the revised manuscript version a sentence about the actually large (millimeter scale) spacing of the STVOs, pointing out that there is no dipolar coupling between the two STVOs: "The coupling is nonlocal and the STVOs are in our case several millimeters apart from each other (however, their distance can be arbitrarily chosen)." in ll. 735 ff.

The reviewer is right that the vortex configuration is the magnetic ground state. In this vortex based STNOs, which we have extensively studied in the last decade, we know that the fundamental mode

corresponding to the spin transfer driven gyrotropic motion of the vortex core is energetically far from other competing modes (see e.g. *Taurel et al., PRB 93, 184427 (2016)*) and hence, a coupling within the MTJ itself can be neglected.

2- It appears to me that there is a distinct operational regime for STVO2 below the sharp claimed threshold around 8 mA, characterized by a broader peak in the frequency spectrum. This regime is not addressed by the authors. The concern is that most of the claimed EP falls within this current range. It is crucial to understand the origin of this signal in the spectrum since it could potentially impact the interpretation of the results. If the different operational regime for STVO2 arises from a distinct mode of oscillation, it could have significant implications for the EP analysis. An EP requires a coalescence of both eigenvalues and eigenvectors (modes). Therefore, if there is a different mode of oscillation within the relevant parameter range, the underlying assumption of coalescing eigenvectors may be invalidated.

My hypothesis is that the different operational regime in STVO2 could be still attributed to a vortex mode with pinning. If this is the case, the effective potential landscape for the magnetization dynamics, which could in turn affect the mode shape. How could this scenario impact the conditions required for an EP in the model.

At dc currents below the critical threshold current for self-sustained oscillations, the STVO is considerably underdamped and hence, no self-sustained oscillations take place neither for the gyrotropic vortex mode nor for any other modes. The reviewer is apparently focusing on the sub-threshold regime for the independent measurements of the STVOs shown in Fig. S3. He/she is right that for STVO2, we do not measure absolutely nothing on the spectrum analyzer as it is the case for the STVO1. Indeed, the existence of a very broad low amplitude thermally excited frequency response is relatively common (see e.g. *R. Lebrun et al, Nat. Comm. 8, 15825 (2017)*). However, its influence can be considered negligible regarding the different orders of magnitude of the frequency response (please note that the graph is in a log scale): The power difference between the self-oscillations (coupled or uncoupled) and the mentioned “broad peak” is up to 3 orders of magnitude as seen in fig. R1 showing the same results in a linear scaling.

Figure R1: Spectra of STVO 2 at different current values. The sub-threshold spectrum is 3 orders of magnitude below the above-threshold, self-sustained, dynamics.

3-The theoretical model is applied with the presumption of the existence of an EP, i.e., a crossing of both eigenvalues and eigenvectors and extract the couplings required for such crossing. Following this assumption, the 'growth or decay rate of the instability', represented as $\text{Re}(iv)$, and the existence of the EP then are both consequence of the chosen coupling. The naïve question about the cause-and-effect relationship between the coupling, the EP, and the instability rate. It would be valuable if the authors could provide a more detailed explanation of their theoretical model, particularly the underlying assumptions and the resulting consequences, would it be for example possible to have same effect of frequency pulling between the oscillators and dragging down of the stability rate to be negative without the existence of the EP.

The presented characteristics are consequences of coupling between the two STVOs. The general approach of our work is to design our experiment in order to achieve the experimental situation close to the compensation of gain and loss. In such case, oscillation death can occur and it is known to appear due to the presence and vicinity of the EP (see also response to point 6). Coupling, EP position and instability are of course linked (as our model describes) and it is remarkable that with the chosen approach of determining the coupling at the EP, all the observed phenomena, e.g. the amplitude death and its parameter dependence, can be modelled in very good agreement. However, as the referee reasonably states also below (after point 6, see also response there), it will be of interest to also thoroughly study the system as a function of the coupling strength (e.g. by adding amplifiers), which would then give a further handling of controlling the EP position and the instability rate. Regarding the existence of an EP, the characteristics of, e.g., amplitude death is a typical non-hermitian effect and an EP always exists in the vicinity, although it might in reality only be "close" and not reachable in the strict sense due to the experimental sensitivity to perturbations around the EP (this fact is indeed exploited for sensor applications). We comment on this in II. 183 ff.: *"Since the condition for the onset of an EP is very sensitive to perturbations, it might happen in experiments that the EP is not reached in a strict sense. Nevertheless, if the parameters are such that the condition (5) is nearly verified, the amplitude death phenomenon is expected to be reliably observed as well."*

4-The authors ascribe the frequency branching, first it is not clear the authors mean by the frequency branching observed in the system, response to the existence of an Exceptional Point (EP). However, if the authors are talking about emergence of additional peaks or the shape (f vs I) this interpretation requires additional justification. It is important to discern and elucidate the specific behaviors or characteristics that can be uniquely attributed to the presence of an EP, and not from coupling such as f vs I . also the presence of intermodulation frequencies is a well-known characteristic of nonlinear coupled oscillators, arising when a system is driven at multiple frequencies or when multiple nonlinear oscillators are coupled. In such cases, the system's response can occur at frequencies that are combinations (sums, differences, multiples) of the original and the driving frequencies. This behavior is intrinsic to nonlinear systems and does not necessarily indicate the presence of an EP. An example from the presented data is where the frequency at 230 MHz can be explained as the difference between the double frequencies 250 minus 270 MHz.

Figure R2: Clarification on modulation effects which we observe in numerical nonlinear simulations and as well in experiment. Modulation is present when two modes exist. However, the effect is unrelated to non-hermitian characteristics.

The emergence of an EP in the coupled system appears naturally, since – as we show – the system is non-hermitian. The EP emerges when the eigenvalues of the 2D system equal. Another characteristics in the vicinity of the EP is the square root like branching (see eq. (3)), which we observe for $I_2 \geq 8$ mA. To be more specific, we write in the revised version (ll. 228 f.): “[...] a square-root-like frequency branching for $I_2 \geq 8$ mA is present.”

As for the second part of the comment, we are aware that modulation indeed exists in the coupled system and, as the reviewer states, it is a well-known phenomenon. However, it is very unrelated to exceptional points and non-hermitian physics. At the frequencies mentioned by the referee, we do not observe any modulation effects and don’t understand from which graph the reviewer has taken those values. In Fig. R2, we try to clearer depict the effect of frequency modulation based on fig. S2 in the manuscript, showing our nonlinear numerical simulations. The case is the same for the experimental data and can also be observed, e.g., in fig. 4d. In the revised version, we add (ll. 688 ff.): “Note that also modulation peaks are visible (such as also visible experimentally, e.g., in fig. 4d), however unrelated to non-hermitian physics discussed here”.

5- regarding the oscillation death Fig 4, it had been shown in the literature previously that injection locking or feedback might very well frustrate the oscillation, also alter the mode shape, (shift the center or make the gyration more elliptical.) how this affects the assumptions of the theory and the availability of overlapped eigen vectors.

First, as described in the text and shown in fig.1, the coupling between the two STVOs is obtained by feeding strip-line antennas above each oscillator with the microwave current I_{rf} generated by the spin torque dynamics occurring in the other STVO. Using this coupling geometry, we believe that none of the mentioned effects take place. Furthermore, reflection that could lead to feedback is reduced by the choice to use devices with a resistance load close to 50 Ohm. Regarding the point raised about the influence of injection locking, we want to point out that no external rf source is used. Moreover, we have utilized STNOs with similar characteristics allowing us to avoid a potential injection locking-like behavior.

6- The simulation part lacks sufficient description of the method used how the two STVOs are placed, the size of the code, and how the coupling form is introduced, also here another opportunity might be done to affirm the effect of EP, by simulating a no crossing STVOs without EP.

The simulation solves the complete nonlinear differential equation of the dynamics (eq. (6)). In the Methods section, we describe all the necessary details, such as the coupling form and the simulation parameters. In order to clarify, in the revised manuscript we refer directly to eq. (6) when describing the simulation approach in the Methods section. Note that the placing of the STVOs is not of importance since our coupling approach does not rely on dipolar interactions and is nonlocal (see response to point 1).

Regarding the affirmation of the EP effect, it is not meaningful to simulate a “no crossing STVO”, since the effect of the coupling would be negligible. Also, an EP would always be present somewhere in the parameter space as long as we have a non-hermitian system. However, the EP might not be easily reachable experimentally. In order to try to convince the reviewer, we present in fig. R3 the situation in which the system eigenvalues are shown for the current value $I_1 = 5$ mA at the same parameters as given in the manuscript. The eigenvalues of both STVOs are very close to those in absence of coupling plotted in black (they almost overlap) and show a substantial change only in the vicinity of the crossing point. In this respect, the graph shows that when the current values of both STVOs are far from their threshold values, measured in absence of coupling, their dynamics is almost the same as when coupled together.

Figure R3: Eigenvalues (real and imaginary parts) for current $I_1 = 5$ mA far from the threshold current. Far from the crossing, the coupling effect is practically negligible.

So, I find the experimental data lacks the convincing breadth, repetition across parameters space, and testing on a number of devices to firmly establish their claims, a counter-example of the same STVOs without an EP and with the presence of similar coupling could bring a contrasting, convincing argument, for example, the coupling parameter change could be done easily adding or removing attenuators, phase shifter, etc.

The manuscript has been under evaluation for quite some time, and unfortunately, during this period, both of the two used STVOs, which are still fragile, MTJ based elements, are no longer functional. As a result, we are unable to provide the requested additional data on these specific devices. However, we do have ongoing experiments with other STVOs that show promising results, and we present some preliminary graphs providing some insights into the behavior of the system under different coupling conditions in fig. R4. Regarding the suggestion of tuning the coupling, we want to emphasize that we have indeed considered this approach. However, this is still a work in

progress and involves a separate set of STVO devices with different frequencies and threshold characteristics. While this may raise doubts about the direct comparison between the two sets of devices, it is essential to clarify that we are exploring the detailed effects of coupling strength in both cases to gain a comprehensive understanding. However, given the time required to gather new data on alternative devices, we think that this is out of the scope of this manuscript and that furthermore, the existing data are convincing enough to provide valuable contributions to the field and support our claims.

Figure R4: Frequency spectra of coupled STVO devices in the vicinity of the EP for different coupling strength, realized by different rf signal amplification. Current $I_1 = 9.7$ mA, magnetic fields $\mu_0 H_{\perp,1} = 435$ mT and $\mu_0 H_{\perp,2} = 370$ mT. Data are preliminary. It can be seen that the branching around 11 mA changes with the coupling, what is expected around the EP.

finally, The current manuscript does not sufficiently clarify the specific ways in which EPs contribute to the functionalization of STVOs, nor does it effectively articulate the advantages of their application, I recommend that the authors rewrite the outlook of their work to provide a clearer picture of the significance and implications of their study rather than the general outlook of spintronic oscillator and EPs.

As for the Outlook section, we have intendedly kept it rather broad, with many references therein (see also our 1st response to reviewer 3 from above) in order to emphasize the system's potential to a broad readership. Indeed, we believe that being more particular on the concrete functionalization, first, goes beyond the scope of the article, and second, would be very specific, contrary to the broad readership of Nature Communications.